# Value-Added White Beer: Influence of Red Grape Skin Extract on the Chemical Composition, Sensory and Antioxidant Properties

Daniela Serea, Georgiana Horincar, Oana Emilia Constantin [ID], Iuliana Aprodu [ID], Nicoleta Stănciuc, Gabriela Elena Bahrim [ID], Silvius Stanciu *[ID] and Gabriela Rapeanu *[ID]

Faculty of Food Science and Engineering, Dunarea de Jos University of Galati, 111 Domneasca Street, 800201 Galati, Romania; daniela.serea@ugal.ro (D.S.); georgiana.parfene@ugal.ro (G.H.); emilia.constantin@ugal.ro (O.E.C.); iuliana.aprodu@ugal.ro (I.A.); nicoleta.stanciuc@ugal.ro (N.S.); gabriela.bahrim@ugal.ro (G.E.B.)

* Correspondence: silvius.stanciu@ugal.ro (S.S.); gabriela.rapeanu@ugal.ro (G.R.)

**Abstract:** This work aimed to improve the functionality of beer by increasing the level of antioxidant activity through the addition, up to acceptable sensory amounts, of red grape skin extract. A commercial hefeweizen beer was supplemented with different concentrations (1, 5, and 10 mg/mL) of grape skin extract (GSE). The phytochemical characterization of GSE and supplemented beer samples was achieved in terms of the total phenolic content (TPC), total flavonoid content (TFC), and total monomeric anthocyanin content (TMA). Additionally, the antioxidant activity of the samples was assessed using a variety of radical scavenging tests. The addition of various concentrations of GSE significantly increased the TPC and TFC content of beer samples, from 3.167 to 4.477 mg GAE/mL and from 0.841 to 1.226 mg CE/mL, respectively. The TMA content of the GSE-supplemented white beer samples ranged from 0.005 to 0.027 mg C3G/ mL. Consequently, the antioxidant capacity of the beer samples increased with the level of GSE addition. The obtained results suggest the potential of using GSE as a functional ingredient for beer production.

**Keywords:** red grape skin extract; beer; phenol compounds; anthocyanin pigments; antioxidant activity

## 1. Introduction

Beer is one of the most consumed alcoholic beverages in the world. This product belongs to a special group of alcoholic beverages with well-defined functional properties. Beer is rich in many endogenous bioactive compounds. Different beer types have various amounts of endogenous antioxidant compounds, depending on the raw materials (malt, non-malted cereal adjuncts, hops) used for brewing and on the particularities of the brewing process. Regarding the raw materials used for brewing, the spectrum and levels of biologically active compounds are usually influenced by genetic and agricultural factors. Phenolic acids, flavonoids, tannins, and amino phenolic compounds are all phenolic substances found in beer [1]. These antioxidants originate from raw materials, such as barley and hop [2]. The phenolic compounds are mainly responsible for the antioxidant activity of beer and play a critical role in preventing oxidation reactions throughout the brewing process and beer storage [3]. In addition, beer contains carbohydrates, amino acids, vitamins, organic acids, bitter substances from hops, and specific compounds with potential beneficial effects for human health if consumed in moderate amounts. Anyway, not all bioactive compounds with antioxidant activity found in beer originate from raw materials. For instance, the Maillard compounds formed during malting and wort boiling are important contributors to the overall antioxidant activity of beer [4]. On the other hand, different processing steps alter the antioxidant activity of the beer, mainly as the result of

reactions involving the polyphenols [5]. It appears that beer filtration, which is meant to ensure the colloidal stability of the products, causes the most significant decrease in the antioxidant activity of beer [6].

Several studies reported that diets rich in bioactive compounds, such as phenolic compounds, are suitable for preventing cardiovascular disease, certain cancers, and other diseases related to aging [7]. Grapes are one of the richest sources of natural polyphenols, among which flavonoids are the most abundant and important for wine quality. Flavonoids are characterized by a 15-carbon structural backbone: C6-C3-C6 (aryl-propyl-aryl). They are typically produced in plants as color pigments and as a defense mechanism in response to environmental changes (exposure to ultra-violet radiation, pathogenic invasion) [8]. According to Constantin et al. [9], the *Băbească neagră* variety is an old local variety of red grapes cultivated in the south-eastern part of Romania at Dealu Bujorului vineyard. This grape variety is mainly used to produce light and fruity wines with 12–12.5% alcohol. Its skin is rich in bioactive compounds that have beneficial effects due to their antioxidant properties.

The aim of the study was to increase the level of bioactive compounds and antioxidant activity of the beer by supplementing it with different concentrations of red grape (*Băbească neagră* variety) skin extract. The stability of the resulting beer samples enriched with bioactive compounds was monitored over 21 days of storage.

## 2. Materials and Methods

### 2.1. Materials

Fresh red grapes (*Băbească neagră* variety) were harvested in September 2020 from the Galati area, Romania. The red grape skins were separated from the pulp, washed with distilled water, placed between filter paper sheets to absorb the excess water, and finally freeze-dried (Christ Alpha 1–4 LD plus, Martin Christ, Osterode am Harz, Germany). The freeze-dried grape skin samples were further ground into powder using a laboratory grinder and kept at 4 °C until analysis. The reagents used in this study were of analytical purity. Delphinidin 3-glucoside, malvidin 3-glucoside, cyanidin 3-glucoside, pelargonidin 3-glucoside, and peonidin chloride, used as standards, were obtained from Extrasynthèse (Z.I Lyon Nord, Genay, France).

### 2.2. Preparation of the Grape Skin Extract (GSE)

The extraction of the bioactive compounds from the lyophilized red grape skins was carried out according to the protocol described by Turturică et al. [10]. Briefly, 1 g of red grape skins powder was mixed with 10 mL of ethanol (96%), and the mixture was placed in an ultrasonic water bath (MRC Instruments, Holon, Israel) at 40 kHz and 50 °C for 55 min. At the end of the ultrasonication treatment, the samples were centrifuged at 5000 rpm and 4 °C for 10 min. The extraction was performed in triplicate. The resulting supernatant volumes were analyzed and then concentrated using a 2–18 stroke concentrator (Christ, UK). The obtained GSE (1 mg) was dissolved in 1 mL of ethanol (96%) and characterized in terms of TPC, TMA, TFC, and antioxidant activity.

### 2.3. Phytochemical Characterization of GSE

2.3.1. Total Monomeric Anthocyanin Content

The TMA content was determined using the pH differential method described by Lee et al. [11], and the results are reported as mg cyanidin-3-glucoside per gram of DW (mg C3G/g DW).

2.3.2. Total Flavonoids Content

The TFC concentration was determined using the method described by Dewanto et al. [12], and the results were expressed as mg catechin equivalents per gram of DW (mg CE / g DW) or milliliter of beer (mg CE / mL) using an equation from the standard catechin calibration curve (y = 2.8919x + 0.006 with $R^2$ = 0.9968).

### 2.3.3. Total Phenolic Content

The TPC was evaluated using the Folin-Ciocalteu spectrophotometric method described by Dewanto et al. [12]. The results were expressed as mg of gallic acid equivalents per gram of DW by means of an equation from the standard gallic acid calibration curve (y = 1.6991x − 0.0256 with $R^2$ = 0.9837).

### 2.3.4. **DPPH Radical Scavenging Activity**

The antioxidant potential of GSE was determined using the DPPH method described by Castro-Vargas et al. [13] and Turturică et al. [10]. Briefly, the reaction mixture was obtained by mixing 200 μL of sample and 3.9 mL of DPPH solution (0.1 M). For 90 min, the mixture was placed in the dark at a temperature of 25 °C. The absorbance of the mixture was measured at the wavelength of 515 nm. The blank was prepared with 200 μL methanol instead of the sample. The data are presented as millimoles of Trolox equivalents per milliliter of GSE in 96% ethanol (mmol TE/mL) and as a percentage of inhibition.

The radical scavenging activity was calculated using the following equation:

$$\% \text{ Inhibition} = (Ab - As)/Ab \times 100) \tag{1}$$

where Ab is the absorbance of the blank sample (distilled water) and As is the absorbance sample.

The equation for the calibration curve of Trolox was y = 0.45x + 0.0075 and $R^2$ = 0.993.

### 2.3.5. ABTS Radical Cation Scavenging Activity

The ABTS scavenger activity of GSE and beer samples was determined using the method described by Zhao et al. [1]. The results are expressed as mmol TE/mL GSE in 96% ethanol and as a percentage of inhibition.

The results were expressed as mmol TE/mL GSE in 96% ethanol or mmol TE/mL beer. The inhibition percentage was calculated using Equation (2):

$$\% \text{ Inhibition} = (Ab - As)/Ab \times 100 \tag{2}$$

where Ab is the absorbance blank (distilled water) and As is the absorbance sample. The equation for the calibration curve of Trolox was y = 0.0045x + 0.0394 and $R^2$ = 0.9889

### 2.3.6. High-Performance Liquid Chromatography (HPLC) Analysis of Anthocyanins

Extract preparation and anthocyanins identification were achieved according to Turturică et al. [10]. The chromatographic profile was determined using a Thermo Finnigan Surveyor HPLC system coupled to a diode array detector (Finnigan Surveyor LC, Thermo Scientific, Waltham, MA, USA). The solvents used were 0.1% formic acid (A) and 100% methanol, and the elution was achieved using the following gradient conditions: 9–35% A (0–20 min); 35%A (20–30 min), 35–50% A (30–40 min), and 50–9% A (40–55 min). The standards and data from the literature were used to achieve anthocyanin quantification [9,14].

### 2.4. Beer Enrichment with GSE and Physicochemical Characterization

A hefeweizen (wheat) beer purchased from a local market of Galati, Romania, was used as a basis for supplementation with GSE such as to obtain a value-added product. Three different concentrations of GSE, namely 1, 5 and 10 mg/mL, were added to the white beer, and the samples were coded B/GSE1, B/GSE5, and B/GSE10, respectively. The highest GSE addition level considered in the study was decided based on the results of a preliminary sensorial test meant to identify the GSE concentration with no impact on the taste of the final product. The control sample was considered the beer without GSE addition. All samples were stored at 4 ± 1 °C, and their quality was evaluated over 21 days of storage at 4 °C in the dark.

The beer samples were evaluated for wort extract, apparent and real extracts, alcohol, and $CO_2$ levels using the beer-3, beer-4, beer-5, and beer-13 methods of ASBC [15].

The CIELAB color parameters (L*, a*, and b*) of the beer samples were determined using a CR300 Chroma Meter (Konica Minolta) coupled with a D65 Illuminant. In addition, the spectrophotometric method was used to measure the color in European Brewery Convention (EBC) units [16].

The pH values were determined directly on the degassed filtered beer samples using a 702SM Titrio pH meter (Metrohm, Herisau, Switzerland).

Phytochemical characterization of the GSE-supplemented beer samples was performed by assessing the antioxidant activity, TMA, TFC, and TPC.

### 2.5. Statistical Analysis

The experiments were performed in triplicate, and the results were expressed as mean ± standard deviation. The differences between the samples were evaluated by the Tukey test with the one-way analysis of variance (ANOVA) method for the data that followed the conditions of normal distribution and equal variations. Minitab 19 software was used to conduct the one-way ANOVA, and Tukey's test with a 95% confidence interval; $p < 0.05$ was statistically significant.

## 3. Results

### 3.1. Phytochemical Characterization of GSE

The phytochemical characterization of GSE was performed by assessing the TMA, TFC, TPC, and antioxidant activity (Table 1). The TPC, TFC, and TMA results obtained for GSE are lower than those obtained by Constantin et al. [17] for the grape skin (*Băbească neagră*). The authors reported a TMA of 20.9 mg C3G/g DW, TFC of 101.10 mg CE/g DW, and TPC of 169.68 mg GAE/g DW. According to Brezoiu et al. [18], the anthocyanin content found in Fetească neagră grape skins was 3.51 ± 0.71 mg C3G/g DW. Regarding the phenolic composition of grape skins, they reported flavonoid and polyphenol contents of 18.96 ± 0.02 and 212.21 ± 0.85, respectively. Guaita and Bosso [19] examined the anthocyanin profile of four skin samples belonging to Albarossa, Barbera, Nebbiolo, and Uvalino grape varieties and reported anthocyanin contents ranging from 9.4 to 21.5 mg C3G/g DW and ABTS antioxidant activity levels of 34.2–51.7 mmol/mL. Yammine et al. [20] reported a TPC of 3.66 ± 0.19 g gallic acid/100 g DW for the Cabernet Franc. Rockenbach et al. [21] characterized the Cabernet Sauvignon, Merlot, Bordeaux, and Isabel grape varieties pomace grown in Brazil and reported a TMA ranging between 1.84 and 11.22 mg malvidin-3-glucoside/g DW.

**Table 1.** Total monomeric anthocyanins (TMA), total flavonoid content (TFC), total phenolic content (TPC), and antioxidant activity (DPPH and ABTS assays) of the grape skins extract (GSE).

| Phytochemical Characteristics | | |
|---|---|---|
| TMA | mg C3G/g DW GSE | 6.26 ± 1.39 |
| TFC | mg CE/g DW GSE | 22.65 ± 0.36 |
| TPC | mg GAE/g DW GSE | 42.44 ± 1.50 |
| **Antioxidant Activity** | | |
| DPPH | Inhibition, % | 63.05 ± 4.37 |
| | mmol TE/mL | 16.50 ± 1.44 |
| ABTS | Inhibition, % | 80.16 ± 0.67 |
| | mmol TE/mL | 1.09 ± 0.01 |

In another study, Santos et al. [22] reported a polyphenolic content of 0.04–122.35 mg GAE/g DW extracted from the pulp and skins of Brazil grape varieties, Benitaka (*Vitis vinifera*) and Isabel and Niagara (*Vitis labrusca*). González-Centeno et al. [23] found in the fresh grapes subjected to ultrasound treatment for 30 min, TPC of 5.37 to 31.87 mg GAE/per 100 g.

### 3.2. HPLC Analysis of Anthocyanins

The HPLC profile of GSE revealed the presence of seven compounds: delphinidin-3-O-glucoside, malvidin 3-O-glucoside, cyanidin 3-O-glucoside, petunidin 3-O-glucoside, pelargonidin 3-O-glucoside, peonidin-3-coumarilglucoside, and peonidin (Figure 1). Budi-Leto et al. [24] obtained similar profiles for 14 grape varieties from the Eastern Adriatic region; the anthocyanins identified were delphinidin, cyanidin, petunidin, peonidin, and malvidin, and their 3-monoglycoside, acetylated, and p-coumaryl derivatives.

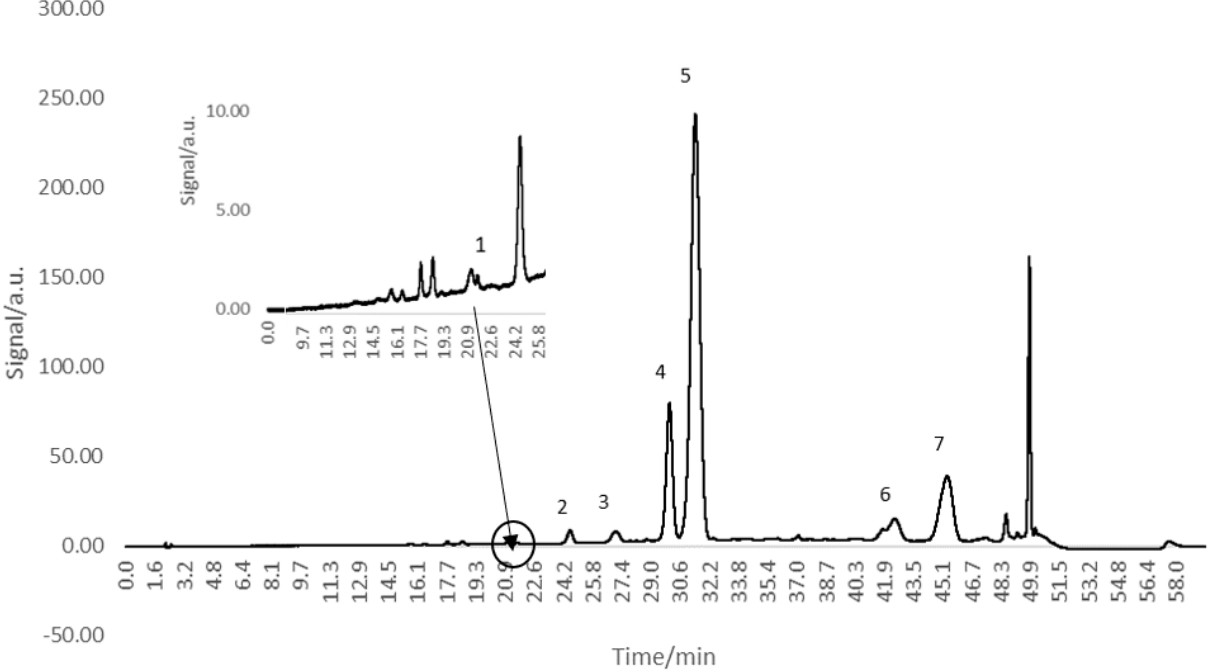

**Figure 1.** HPLC chromatograms of anthocyanin/anthocyanidin profile of *Băbească neagră* grape skin at 520 nm: 1-delphinidin 3-O-glucoside, 2-cyanidin 3-O-glucoside, 3-petunidin 3-O-glucoside, 4-pelargonidin 3-O-glucoside, 5-malvidin-3-O-glucoside, 6-peonidin-3-coumarilglucoside, and 7-peonidin.

Malvidin 3-O-glucoside is the major compound found in the GSE with a 10.92 $\pm$ 0.00 mg/g DW concentration. Similar findings were reported by Benmeziane et al. [25] for Gros noir and Muscat noir grape varieties, where malvidin 3-O-glucoside was the major anthocyanin component. The same primary compound was identified by Kharadze et al. [26] in five red grape varieties (Alexandrouli, Mujuretuli, Saperavi, Otskhanuri, Sapere, and Ojaleshi). Additionally, Budi-Leto et al. [24] found 14 anthocyanins in different grape varieties, with malvidin 3-monoglucoside prevailing in the mixture in the case of G1, IJK 92, and 'Merlot' samples. Silva and Queiroz [27] revealed that the primary anthocyanin found in Touriga Nacional red grapes from the Dao area (Portugal) was malvidin-3-O-glucoside.

### 3.3. Phytochemical Characterization of Beer Enriched with GSE

Increasing the nutritional value of food products and beverages can be achieved by enriching the content of the bioactive compounds through the addition of extracts, fruits, or fruit juices. The suitability of the GSE to be used for enriching the phytochemical profile of different beverages was tested by supplementing beer samples with different concentrations of the extract (1, 5, and 10 mg GSE/mL). The physico-chemical characteristics of the beer are shown in Table 2.

**Table 2.** Physico-chemical parameters of the control of beer (without GSE).

| Physico-Chemical Characteristics | Beer Control |
|---|---|
| Alcohol, % mass | $4.29 \pm 0.01$ |
| Alcohol, % vol | $5.39 \pm 0.02$ |
| Real extract, °P | $4.25 \pm 0.01$ |
| Original extract, °P | $12.48 \pm 0.02$ |
| Apparent extract, °P | $2.05 \pm 0.01$ |
| $CO_2$, g/100 mL | $0.68 \pm 0.02$ |
| pH | $4.88 \pm 0.02$ |

Beer supplementation with GSE resulted in no significant change in the main physico-chemical characteristics of the control samples. Similar findings have been reported by Ulloa et al. [2] for beer supplemented with propolis extract and by Horincar et al. [28], who supplemented lager beer with eggplant peel extract. They stated that there were no changes in the physicochemical characteristics of the beer when the phytochemical extracts were added.

The level of bioactive compounds of the GSE-supplemented beer samples was assessed by determining the TPC, TMA, and TFC (Table 3). The phytochemical composition of the beer samples was monitored throughout 21-days of storage in refrigerated conditions.

**Table 3.** Phytochemical characterization of control and value-added beer enriched with grape skin extract (GSE): BC—control beer; B/GSE1—beer with 1 mg GSE/mL; B/GSE 5—beer with 5 mg GSE/mL and B/GSE 10—beer with 10 mg GSE/mL, over 21 days of storage.

| Sample | Bioactive Compounds | Storage Time (Days) | | | |
|---|---|---|---|---|---|
| | | 0 | 7 | 14 | 21 |
| BC | TMA, mgC3G/mL | Nd. * | Nd. * | Nd. * | Nd. * |
| | TFC, mg CE/mL | $0.841 \pm 0.004$ [aD] | $0.808 \pm 0.010$ [aC] | $0.786 \pm 0.077$ [aC] | $0.788 \pm 0.058$ [aB] |
| | TPC, mg GAE/mL | $3.167 \pm 0.059$ [aC] | $3.118 \pm 0.228$ [aC] | $3.080 \pm 0.066$ [aC] | $2.975 \pm 0.064$ [aC] |
| B/GSE1 | TMA, mgC3G/mL | $0.005 \pm 0.000$ [aC] | $0.005 \pm 0.000$ [aC] | $0.005 \pm 0.000$ [aC] | $0.004 \pm 0.001$ [aC] |
| | TFC, mg CE/mL | $0.964 \pm 0.047$ [aC] | $0.961 \pm 0.105$ [aBC] | $0.961 \pm 0.105$ [aBC] | $0.919 \pm 0.118$ [aAB] |
| | TPC, mg GAE/mL | $3.640 \pm 0.299$ [aB] | $3.640 \pm 0.061$ [aB] | $3.530 \pm 0.233$ [aBC] | $3.375 \pm 0.171$ [aB] |
| B/GSE5 | TMA, mgC3G/mL | $0.019 \pm 0.000$ [aB] | $0.017 \pm 0.000$ [bB] | $0.016 \pm 0.001$ [cB] | $0.016 \pm 0.001$ [cB] |
| | TFC, mg CE/mL | $1.096 \pm 0.039$ [aB] | $1.071 \pm 0.091$ [aAB] | $1.063 \pm 0.085$ [aAB] | $1.009 \pm 0.020$ [aAB] |
| | TPC, mg GAE/mL | $3.995 \pm 0.096$ [aA] | $3.981 \pm 0.188$ [aB] | $3.804 \pm 0.435$ [aAB] | $3.608 \pm 0.145$ [aB] |
| B/GSE10 | TMA, mgC3G/mL | $0.027 \pm 0.001$ [aA] | $0.027 \pm 0.001$ [aA] | $0.023 \pm 0.001$ [BA] | $0.022 \pm 0.001$ [bA] |
| | TFC, mg CE/mL | $1.226 \pm 0.029$ [aA] | $1.208 \pm 0.038$ [aA] | $1.176 \pm 0.048$ [aA] | $1.096 \pm 0.141$ [aA] |
| | TPC, mg GAE/mL | $4.477 \pm 0.101$ [aA] | $4.469 \pm 0.060$ [aA] | $4.210 \pm 0.113$ [aA] | $4.057 \pm 0.066$ [bA] |

* Nd.—concentration not detected; Mean values followed by different lowercase letters ([a–c]) in the same row and different uppercase letters ([A–C]) in the same column are statistically different, based on the Tukey method and 95% confidence.

The results presented in Table 3 indicate that the concentration of TMA increases with the amount of GSE added to the beer. In the present study, beer enriched with the three concentrations of GSE recorded a TMA content that ranged from 0.005 to 0.027 mg C3G/mL (Table 3). Regarding the stability of the TMA content over the storage period, a slight decrease ($p < 0.05$) was observed in the case of the beer samples supplemented with higher levels of GSE (B/GSE5 and B/GSE10). The susceptibility to degradation of the anthocyanins under different storage conditions, such as pH, exposure to light, oxygen access, and temperature, might explain the reduction of the TMA levels in the investigated beer samples.

The control beer had a TFC of 0.841 mg CE/mL and a TPC of 3.167 mg GAE/mL, which were relatively stable throughout the storage period (Table 3). The high TPC is due to the fact that a hefeweizen beer type was used in the present study for supplementation with

biologically active compounds. A wide spectrum of phenolic compounds with antioxidant activity was identified in beer. The usual phenolic profile of beer includes phenolic acids, flavonoids, proanthocyanins, tannins, and amino phenolic compounds [1]. The level of phenolic compounds found in different types of beer varies with the growing conditions and varieties of barley and hops, as well as with brewing parameters. In addition to contributing to the antioxidant activity of the beer, these compounds influence important attributes of the beers, such as the color, bitterness, astringency, flavor, colloidal stability, and shelf life [16,29].

As expected, GSE addition to the beer sample resulted in the gradual increase in both TFC and TPC. Regardless of the GSE concentration, the value-added beer samples exhibited rather good stability of both TFC and TPC (Table 3).

Our findings regarding the TFC and TPC of the value-added beer samples are in overall good agreement with the literature. Dorđević et al. [30] found that the total phenols content was highest in lager beer supplemented with thyme, juniper, and lemon balm (384.22, 365.38, and 363.08 mg GAE/L, respectively), representing an increase of 37.09%, 30.36%, and 29.55%, respectively, compared to the commercial lager beer. Horincar et al. [28] claimed that the flavonoid content of the beer samples enriched with eggplant (Solanum melongena L.) extract varied from 0.07 to 0.17 mg CE/mL, while the polyphenol content varied from 0.43 to 0.63 mg GAE/mL. Obtaining beer with hawthorn juice and hawthorn fruit was the aim of the study performed by Gasiński et al. [31], who reported TPC of 0.27 and 0.41 mg GAE/mL, respectively. Ducruet et al. [32] added goji berries to the beginning of wort boiling and obtained a special ale with pleasant sensory characteristics, high antioxidant capacity, and total polyphenol content of 0.62 mg GAE/mL.

### 3.4. Antioxidant Activity of Beer Enriched with GSE

Different studies indicate that the antioxidant activity of beer is mainly due to the phenolic compounds originating from the main ingredient used for producing the beer and to the Maillard compounds present in malt or formed while boiling the wort [5,16]. The contribution of the trace compounds or elements to the antioxidant capacity of beer, as well as the synergies of different biologically active compounds, should also be factored in [5]. In Figure 2, the results showing the antioxidant activity of the beer samples considered in the study are presented.

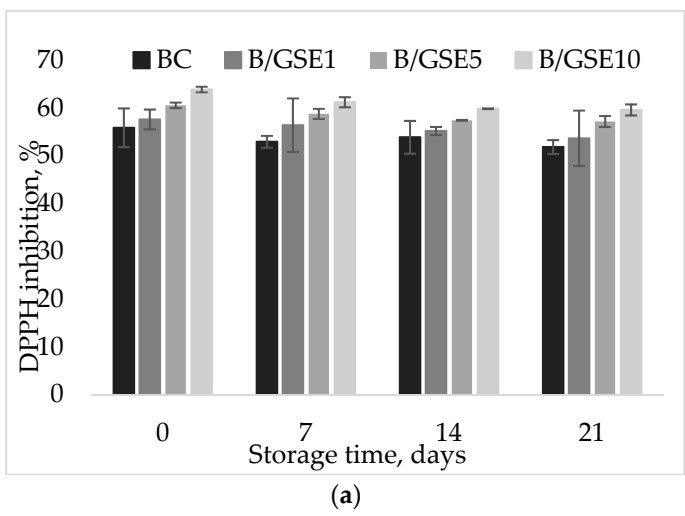
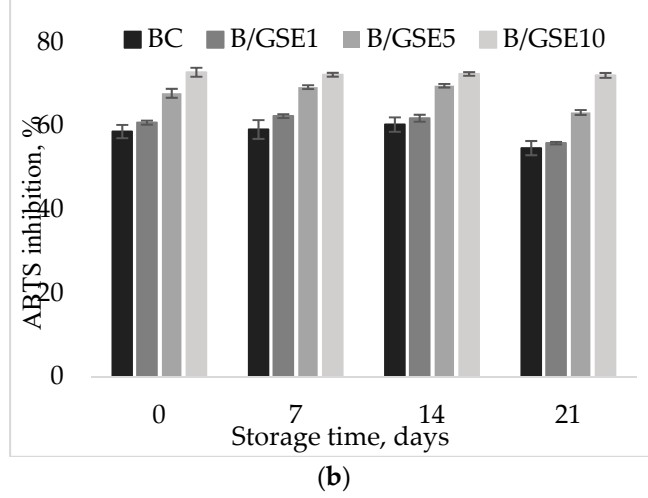

(**a**) (**b**)

**Figure 2.** Radical scavenging activity of DPPH (**a**) and ABTS (**b**) exerted by the valued-added beer supplemented with grape skin extract (GSE): BC—control beer; B/GSE1—beer with 1 mg GSE/mL; B/GSE 5—beer with 5 mg GSE/mL and B/GSE 10—beer with 10 mg GSE/mL, over 21 days of storage.

As a result of increasing the level of different kinds of bioactive compounds, the antioxidant activity of the beer samples increases with the added level of GSE (Figure 2).

The antioxidant potential of the beer samples was determined by quantifying the free radical scavenging activity of DPPH and ABTS, and both methods indicated overall good stability during storage. A similar trend was previously noted by Horincar et al. [28] and Đorđevic et al. [30], who used different levels of various vegetable extracts to boost the biological activity of the lager beer.

According to Horincar et al. [28], at the end of 21 days of storage, the DPPH radical scavenging activity of the lager beer supplemented with different amounts of eggplant peel extract (1–10 mg EPE/mL) was significantly higher (78.04–79.15%) compared to the control sample (56.41%). Similar findings were provided by the ABTS-based method; Horincar et al. [28] reported the highest antioxidant activity of 0.140 mmol TE/mL and inhibition of 80.019% for the sample supplemented with 10 mg EPE/mL, whereas the control sample exhibited an antioxidant activity of 0.090 mmol TE/mL, and inhibition of 57.288%. Similarly, Ulloa et al. [2] reported antioxidant activity of 0.014–0.044 mmol TE/mL when using the DPPH free radical method and of 0.079–0.149 mmol TE/mL when using the ABTS free radical method for the beer samples supplemented with various amounts of propolis extract (0.05, 0.15, and 0.25 g/L). Higher antioxidant activity of 0.443–2.175 mmol TE /mL in the case of the DPPH method and of 1.356–2.041 mmol TE/mL in the case of the ABTS method were found by Gasiński et al. [31], who characterized beer samples supplemented with hawthorn fruit and hawthorn juice.

### 3.5. Color of Beer Enriched with GSE

The color characteristics of the beer with different levels of GSE were classified using the recommended EBC method and the CIELAB color system. The visual sense contains three types of red, green, and blue cone receptors, existing in unequal quantities so that the color can be perceived differently [33]. The color of beer mainly depends on the melanoidins and caramel from malt and adjuncts and on the processing parameters, mainly the intensity of the wort boiling [34]. The oxidized polyphenols might also contribute significantly to the beer [35].

GSE addition to beer resulted in significant changes in the color parameters (Figure 3). The impact of GSE addition on beer color was monitored during the storage period of 21 days (Table 4), and the CIELAB analysis proved to be more effective in differentiating between beer samples compared to the EBC method. The significant decrease in the lightness of the samples and the increase in the a* values were noticed when raising the concentration of added GSE ($p < 0.05$). Our results are in good agreement with Horincar et al. [28]. A significant increase of the EBC unit from 8.28 to 22.65 at the end of the tested storage period was also noticed upon increasing the concentration of added GSE.

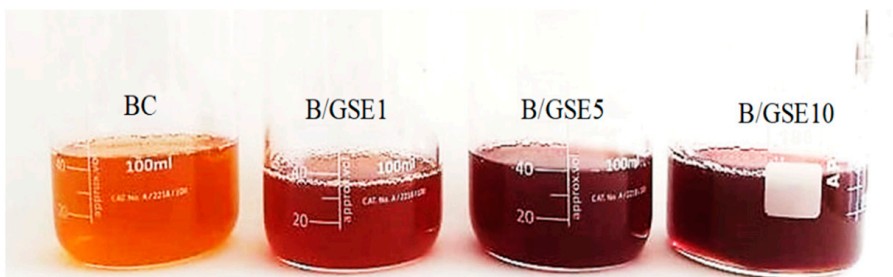

**Figure 3.** The beer samples obtained by adding different concentrations of grape skin extract (GSE): BC—control beer; B/GSE1—beer with 1 mg DSE/mL; B/GSE 5—beer with 5 mg GSE/mL and B/GSE 10—beer with 10 mg GSE/mL.

The color parameters varied during the storage period, regardless of the beer sample investigated. The most important changes were noticed in the case of the beer supplemented with 10 mg/mL (B/GSE 10). The L* value increased from 40.28 to 45.89 after 21 days of storage ($p < 0.05$). Over the entire period considered for the stability test, an in-

tense reddish hue was measured in the sample, decreasing from 25.81 to 21.48. Taking into account that the flavylium cation of anthocyanins is usually responsible for the purple and red color, we might consider that the redness decrease during storage occurs as the result of the rather high instability of the anthocyanins [36]. Finally, there were no significant changes in the yellow (b*) contribution to the overall color of the control beer and samples supplemented with different amounts of GSE during storage.

**Table 4.** Evolution of the color parameters of the control and value-added beer enriched with grape skin extract (GSE): BC—control beer; B/GSE1—beer with 1 mg GSE/mL; B/GSE 5—beer with 5 mg GSE/mL; and B/GSE 10—beer with 10 mg GSE/mL, over 21 days of storage.

| Sample | Color Parameters | Storage Time (Days) | | | |
|---|---|---|---|---|---|
| | | 0 | 7 | 14 | 21 |
| BC | L* | 66.15 ± 0.11 [bA] | 66.39 ± 0.27 [bA] | 66.88 ± 0.36 [abA] | 67.83 ± 0.81 [aA] |
| | a* | 0.89 ± 0.07 [aD] | 0.92 ± 0.03 [aD] | 0.88 ± 0.07 [aD] | 0.88 ± 0.01 [aD] |
| | b* | 10.36 ± 0.47 [aA] | 11.46 ± 0.53 [aA] | 11.01 ± 0.13 [aA] | 10.00 ± 1.02 [aAB] |
| | EBC | 8.28 ± 0.26 [aD] | 8.29 ± 0.25 [aD] | 8.28 ± 0.27 [aD] | 8.28 ± 0.27 [aD] |
| B/GSE 1 | L* | 62.93 ± 0.20 [aB] | 63.65 ± 0.48 [aB] | 63.35 ± 0.14 [aB] | 63.15 ± 0.76 [aB] |
| | a* | 4.78 ± 0.09 [aC] | 4.63 ± 0.33 [aC] | 4.71 ± 0.20 [aC] | 4.92 ± 0.39 [aC] |
| | b* | 10.66 ± 0.24 [aA] | 9.52 ± 0.32 [abB] | 9.32 ± 0.49 [bC] | 9.63 ± 0.74 [abAB] |
| | EBC | 11.27 ± 0.22 [aC] | 11.23 ± 0.22 [aC] | 11.19 ± 0.28 [aC] | 11.27 ± 0.22 [aC] |
| B/GSE 5 | L* | 47.87 ± 0.02 [bC] | 49.76 ± 0.78 [aC] | 48.22 ± 0.06 [bC] | 48.01 ± 0.33 [bC] |
| | a* | 17.41 ± 0.01 [aB] | 15.67 ± 0.41 [cB] | 16.13 ± 0.14 [bcB] | 16.50 ± 0.24 [bB] |
| | b* | 10.48 ± 0.05 [bA] | 10.09 ± 0.16 [cB] | 10.33 ± 0.04 [bcB] | 11.34 ± 0.17 [aA] |
| | EBC | 14.44 ± 0.20 [aB] | 14.42 ± 0.23 [aB] | 14.42 ± 0.26 [aB] | 14.44 ± 0.20 [aB] |
| B/GSE 10 | L* | 40.28 ± 0.01 [bD] | 42.53 ± 0.27 [bD] | 42.94 ± 0.09 [abD] | 45.89 ± 2.52 [aC] |
| | a* | 25.81 ± 0.05 [aA] | 23.86 ± 0.07 [bA] | 23.83 ± 0.11 [bA] | 21.48 ± 1.27 [cA] |
| | b* | 8.91 ± 0.03 [aB] | 8.44 ± 0.04 [aC] | 8.57 ± 0.04 [aD] | 8.65 ± 0.53 [aB] |
| | EBC | 22.65 ± 0.20 [aA] | 22.66 ± 0.18 [aA] | 22.66 ± 0.16 [aA] | 22.65 ± 0.20 [aA] |

L*—lightness; a*—green-to-red; b*—blue-to-yellow; For each color parameter, mean values followed by different uppercase letters ([A–D]) on the same column and by different lowercase letters ([a–c]) on the same row are statistically different $p < 0.05$.

## 4. Conclusions

The study indicates that the use of GSE in beer leads to an increase in the level of bioactive compounds and the antioxidant potential of beer. GSE added to beer has increased the phenolic compound content, which is usually influenced by the brewing process. In addition to these compounds, the addition of GSE provided significant amounts of anthocyanins to the beer samples. The evolution of bioactive compounds of the SGE-supplemented beer during storage was as follows: the anthocyanin content decreased over time, while the amounts of flavonoids showed fairly good stability. For these reasons, we can conclude that beer supplemented with GSE has a high functional potential. The added values were achieved by using plant raw materials rich in bioactive compounds and not from artificial additives. Moreover, the plant material used can have multiple uses in the food industry and the production of value-added products and can help reduce waste by implementing an economic model of environmental protection. The study will be further developed, aiming at monitoring the behavior of beer over the typical period of storage. Moreover, the sensory evaluation of beer samples supplemented with GSE will be considered.

**Author Contributions:** Conceptualization, D.S., G.H., I.A. and G.R.; methodology, D.S. and I.A.; software O.E.C.; validation, N.S. and G.R.; formal analysis, D.S. and I.A.; investigation, D.S. and I.A.; resources, G.E.B. and G.R.; data curation, D.S. and G.H.; writing—original draft preparation, D.S., G.H. and I.A.; writing—review and editing, G.R. and I.A.; visualization, N.S.; supervision, G.E.B.;

project administration, N.S.; funding acquisition, G.E.B., S.S. and G.R. All authors have read and agreed to the published version of the manuscript.

**Funding:** This work was supported by the Internal Grant financed by Dunarea de Jos University of Galati, Romania, contract no.14888/11.05.2022 (CNFIS-FDI-2022-0205).

**Institutional Review Board Statement:** Not applicable.

**Informed Consent Statement:** Not applicable.

**Data Availability Statement:** The data that support the findings of this study are available from the corresponding author, G.R., upon reasonable request.

**Acknowledgments:** The Integrated Center for Research, Expertise, and Technological Transfer in the Food Industry is acknowledged for providing technical support.

**Conflicts of Interest:** The authors declare no conflict of interest.

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
