# Peer review of "Value-Added White Beer: Influence of Red Grape Skin Extract on the Chemical Composition, Sensory and Antioxidant Properties"

_sustainability, doi:10.3390/su14159040_

Round 1

Reviewer 1 Report

The paper relates the addition of grape skin extracts to white beer and the effect on some chemical parameters. In my opinion, the research lack in originality and only very marginally contributes to the advancement of brewing sector. The main question to be investigated by the authors was: “Can a phenolic rich extract increase the phenolic content of a beer?” the obvious answer was: “Yes”. But apart from this (quite expected) result, nothing more was given by the paper. In addition, the paper has a number of weakness which make it not publishable, in my opinion: 1) the experimental plan is not well explained; 2) the authors did not take into consideration the effects of the opening of the bottles (or the tanks) into the shelf life simulation; 3) Storage was monitored for 20 day at 4°C . This is not a “storage” time relevant with the shelf life of a beer. Due to this is not surprising that poor changes in were found in phenolics at the end of this period of time. 4) Nothing is known about the effect of the addition of such a huge amount of phenolics (up to 10 g/L) from the sensory point of view (taste, acidity, foam etc). Is this addition acceptable for the consumer? 5) Anthocyanins analysis gave some unexpected results (lack of delphinidin or acetylated and coumarylated anthocyanins, but the presence of peonidin---coumarylglucoside) very uncommon as anthocyanin profile.

Other comments follow:

Line 41: Please give a reference for the statement on antioxidant activity of Maillard products

Line 78: Not clear here: were the analysis carried out on the joined surnatant (30mL) or on the concentrated one (1 mL)?

Line 80: please amend the sentence

Lines 113-115: Not clear here: 1) Were the beers purchased once bottled? 2) Then the author had to open the beers and add the GSE giving rise to oxygenation and loosing CO2? 3) how and where the added beer stored? (Dark?, in the same bottle?, with saturated headspace?)

Line 145: Did your phenolic content is given as DW as the ones from Constantin? If yes, this is not stated in mat and meth and would partially justify the differences.

Line 166: No delphinidin?

Line 171: ? did you mean p-coumaryl?

Figure 1: Please, flag each peak with the name or indicate in the caption the name corresponding to each peak.

Line 189: 1,5 and 10 mg GSE each mL of beer? Which would mean up to 10 g/L of beer?

Line 193: This is quite surprising since the addition of 10g/L of extract should impact the extract and the pH. Could you fournish the data?

Table 4: The amounts of TMA are somewhat unconvincing. In B/GSE1 the authors added 1g/L extract (e.g. 6.26 mg of TMA according to table 1). Then why they find 60 mg/L in the added beer?. The same for all the other: In B/GSE10 they added 10 g/L GSe which means 62.6 mg TMA but in the final beer TMA were 320 mg/L at time 0. How the author could justify?

Line 300-301: What kind of comparison is this? Why to compare White beer added of GSE with Ale beer added with unknown amounts of Hibiscus?

Reviewer 2 Report

The calibration curves equations of the used methods (TPC, DPPH radical scavenging activity, ABTS scavenger activity) must be  added

The HPLC work parameters, the employed standards as well as the equipment must be mentioned

Figure 1 has uncomplete legend 

Reviewer 3 Report

Very nice work. Congratulations!

Reviewer 4 Report

The results obtained in this study show that the addition of GSE in beer lead to an increase in the level of bioactive compounds and the antioxidant potential of beer. GSE added to beer has increased the phenolic compound concentration. Also, the addition GSE provided significant concentration of anthocyanins in beer. 

A sensory evaluation of beer samples with the addition of GSE would be a beneficial to the manuscript quality. In my opinion it is of importance for a new product such as this. It is not mandatory, but if it is possible please add a beer sensory evaluation since the addition of GSE most certainly influenced beer taste and aroma.  

Please correct the following:

Table 3 - Instead of Primitive extract should be Original extract

Line 223 - instead of was identifies should be was identified 

Line 295 - instead of ECB method should be EBC method

Round 2

Reviewer 1 Report

The paper is, in my opinion, still unconvincing. I appreciated a lot the responses given by the authors, but the overall figures of merit of the work is, in my opinion, not sufficient.

The authors wanted to investigate on the addition of grape extracts to beer, to produce an enriched beverage but if the resulting beer is not sensorially acceptable, then the work is useless.

Further:

1)      How could the authors identify petunidin-3G and petunidin-3-coumaroylglucoside without standard compounds (see paragraph 2.1)?

2)      Table 4, as presented on the amended PDF is almost entirely unreadable, hence I could not verify the changes done in the quantification
